# Norm-Ranging LSH for Maximum Inner Product Search

**Xiao Yan, Jinfeng Li, Xinyan Dai, Hongzhi Chen, James Cheng**
Department of Computer Science
The Chinese University of Hong Kong
Shatin, Hong Kong
{xyan, jfli, xydai, hzchen, jcheng}@cse.cuhk.edu.hk

## Abstract

Neyshabur and Srebro proposed SIMPLE-LSH [2015], which is the state-of-the-art hashing based algorithm for maximum inner product search (MIPS). We found that the performance of SIMPLE-LSH, in both theory and practice, suffers from long tails in the 2-norm distribution of real datasets. We propose NORM-RANGING LSH, which addresses the excessive normalization problem caused by long tails by partitioning a dataset into sub-datasets and building a hash index for each sub-dataset independently. We prove that NORM-RANGING LSH achieves lower query time complexity than SIMPLE-LSH under mild conditions. We also show that the idea of dataset partitioning can improve another hashing based MIPS algorithm. Experiments show that NORM-RANGING LSH probes much less items than SIMPLE-LSH at the same recall, thus significantly benefiting MIPS based applications.

## 1 Introduction

Given a dataset $\mathcal{S} \subset \mathbb{R}^d$ containing $n$ vectors (also called items) and a query $q \in \mathbb{R}^d$, maximum inner product search (MIPS) finds the vector in $\mathcal{S}$ that has the maximum inner product with $q$,

$$p = \arg\max_{x \in \mathcal{S}} q^\top x. \tag{1}$$

MIPS may require items with the top $k$ inner products and it usually suffices to return approximate results (i.e., items with inner products close to the maximum). MIPS has many important applications including recommendation based on user and item embeddings obtained from matrix factorization [Koren et al., 2009], multi-class classification with linear classifier [Dean et al., 2013], filtering in computer vision [Felzenszwalb et al., 2010], etc.

MIPS is a challenging problem as modern datasets often have high dimensionality and large cardinality. Initially, tree-based methods [Ram and Gray, 2012, Koenigstein et al., 2012] were proposed for MIPS, which use the idea of branch and bound similar to k-d tree [Friedman and Tukey, 1974]. However, these methods suffer from the curse of dimensionality and their performance can be even worse than linear scan when feature dimension is as low as 20 [Weber et al., 1998]. Shrivastava and Li proposed L2-ALSH [2014], which attains the first provable sub-linear query time complexity for approximate MIPS that is independent of dimensionality. L2-ALSH applies an asymmetric transformation [1] to transform MIPS into $L_2$ similarity search, which can be solved with well-known LSH functions. Following the idea of L2-ALSH, Shrivastava and Li formulated another pair of asymmetric transformations called SIGN-ALSH [2015] to transform MIPS into angular similarity search and obtained lower query time complexity than that of L2-ALSH.

Neyshabur and Srebro showed that asymmetry is not necessary when queries are normalized and items have bounded 2-norm [2015]. They proposed SIMPLE-LSH, which adopts a symmetric transformation and transforms MIPS into angular similarity search similar to SIGN-ALSH. Moreover, they proved that SIMPLE-LSH is a universal LSH for MIPS, while L2-ALSH and SIGN-ALSH are not. SIMPLE-LSH is also parameter-free and avoids the parameter tuning of L2-ALSH and SIGN-ALSH. Most importantly, SIMPLE-LSH achieves superior performance over L2-ALSH and SIGN-ALSH in both theory and practice, and thus is the state-of-the-art hashing based algorithm for MIPS.

SIMPLE-LSH requires the 2-norms of the items to be bounded, which is achieved by normalizing the items with the maximum 2-norm in the dataset. However, real datasets often have long tails in the distribution of 2-norm, meaning that the maximum 2-norm can be much larger than the majority of the items. As we will show in Section 3.1, the excessive normalization of SIMPLE-LSH makes the maximum inner product between the query and the items small, which harms the performance of SIMPLE-LSH in both theory and practice.

To solve this problem, we propose NORM-RANGING LSH. The idea is to partition the dataset into multiple sub-datasets according to the percentiles of the 2-norm distribution. For each sub-dataset, NORM-RANGING LSH uses SIMPLE-LSH as a subroutine to build an index independent of other sub-datasets. As each sub-dataset is normalized by its own maximum 2-norm, which is usually significantly smaller than the maximum 2-norm in the entire dataset, NORM-RANGING LSH achieves lower query time complexity than SIMPLE-LSH. To support efficient query processing, we also formulate a similarity metric which defines a probing order for buckets from different sub-datasets. We compare NORM-RANGING LSH with SIMPLE-LSH and L2-ALSH on three real datasets and show empirically that NORM-RANGING LSH offers up to an order of magnitude speedup.

## 2  Locality Sensitive Hashing for MIPS

### 2.1  Locality Sensitive Hashing

A definition of locality sensitive hashing (LSH) [Indyk and Motwani, 1998, Andoni et al., 2018] is given as follows:

**Definition 1.** *(LSH) A family $\mathcal{H}$ is called $(S_0, cS_0, p_1, p_2)$-sensitive if, for any two vectors $x, y \in \mathbb{R}^d$:*

- *if $sim(x, y) \geq S_0$, then $\mathbb{P}_{\mathcal{H}}[h(x) = h(y)] \geq p_1$,*
- *if $sim(x, y) \leq cS_0$, then $\mathbb{P}_{\mathcal{H}}[h(x) = h(y)] \leq p_2$.*

Note the original LSH is defined for distance function, we adopt a formalization adapted for similarity function [Shrivastava and Li, 2014], which is more suitable for MIPS. For a family of LSH to be useful, it is required that $p_1 > p_2$ and $0 < c < 1$. Given a family of $(S_0, cS_0, p_1, p_2)$-LSH, a query for $c$-approximate nearest neighbor search [2] can be processed with a time complexity of $\mathcal{O}(n^\rho \log n)$, where $\rho = \frac{\log p_1}{\log p_2}$. For $L_2$ distance, there exists a well-known family of LSH defined as follows:

$$h_{a,b}^{L_2}(x) = \left\lfloor \frac{a^\top x + b}{r} \right\rfloor, \tag{2}$$

where $\lfloor \rfloor$ is the floor operation, $a$ is a random vector whose entries follow i.i.d. standard normal distribution and $b$ is generated by a uniform distribution over $[0, r]$. When a hash function is drawn randomly and independently for each pair of vectors [Wang et al., 2013], the collision probability of (2) is given as:

$$\mathbb{P}_{\mathcal{H}}\left[h_{a,b}^{L_2}(x) = h_{a,b}^{L_2}(y)\right] = F_r(d) = 1 - 2\Phi(-\frac{r}{d}) - \frac{2d}{\sqrt{2\pi}r}(1 - e^{-(r/d)^2/2}), \tag{3}$$

in which $\Phi(x)$ is the cumulative density function of standard normal distribution and $d = \|x - y\|$ is the $L_2$ distance between $x$ and $y$. For angular similarity, sign random projection is an LSH. Its expression and collision probability can be given as [Goemans and Williamson, 1995]:

$$h_a(x) = \text{sign}(a^\top x), \mathbb{P}_{\mathcal{H}}[h_a(x) = h_a(y)] = 1 - \frac{1}{\pi}\cos^{-1}\left(\frac{x^\top y}{\|x\|\|y\|}\right), \tag{4}$$

where the entries of $a$ follow i.i.d. standard normal distribution.

## 2.2 L2-ALSH

Shrivastava and Li proved that there exists no symmetric LSH for MIPS if the domain of the item $x$ and query $q$ are both $\mathbb{R}^d$ [2014]. They applied a pair of asymmetric transformations, $P(x)$ and $Q(q)$, to the items and the query, respectively.

$$P(x) = [Ux; \|Ux\|^2; \|Ux\|^4; ...; \|Ux\|^{2^m}]; \quad Q(q) = [q; 1/2; 1/2; ...; 1/2] \tag{5}$$

The scaling factor $U$ should ensure that $\|Ux\| < 1$ for all $x \in \mathcal{S}$ and the query is normalized to unit 2-norm before the transformation. After the transformation, we have:

$$\|P(x) - Q(q)\|^2 = 1 + \frac{m}{4} - 2Ux^\top q + \|Ux\|^{2^{m+1}}. \tag{6}$$

As the scaling factor $U$ is common for all items and the last term vanishes with sufficiently large $m$ because $\|Ux\| < 1$, (6) shows that the problem of MIPS is transformed into finding the nearest neighbor of $Q(q)$ in terms of $L_2$ distance, which can be solved using the hash function in (2). Given $S_0$ and $c$, a query time complexity of $\mathcal{O}(n^\rho \log n)$ can be obtained for $c$-approximate MIPS with:

$$\rho = \frac{\log F_r(\sqrt{1 + m/4 - 2US_0 + (US_0)^{2^{m+1}}})}{\log F_r(\sqrt{1 + m/4 - 2cUS_0})}. \tag{7}$$

It is suggested to use a grid search to find the parameters ($m$, $U$ and $r$) that minimize $\rho$.

## 2.3 SIMPLE-LSH

Neyshabur and Srebro proved that L2-ALSH is not a universal LSH for MIPS, that is [2015], for any setting of $m$, $U$ and $r$, there always exists a pair of $S_0$ and $c$ such that $x^\top q = S_0$ and $y^\top q = cS_0$ but $\mathbb{P}_{\mathcal{H}}[h_{a,b}^{L_2}(P(x)) = h_{a,b}^{L_2}(Q(q))] < \mathbb{P}_{\mathcal{H}}[h_{a,b}^{L_2}(P(y)) = h_{a,b}^{L_2}(Q(q))]$. Moreover, they showed that asymmetry is not necessary if the items have bounded 2-norm and the query is normalized, which is exactly the assumption of L2-ALSH. They proposed a symmetric transformation to transform MIPS into angular similarity search as follows:

$$P(x) = [x; \sqrt{1 - \|x\|^2}]; P(q)^\top P(x) = [q; 0]^\top [x; \sqrt{1 - \|x\|^2}] = q^\top x. \tag{8}$$

They apply the sign random projection in (4) to $P(x)$ and $P(q)$ to obtain an LSH for $c$-approximate MIPS with a query time complexity $\mathcal{O}(n^\rho \log n)$ and $\rho$ is given as:

$$\rho = G(c, S_0) = \frac{\log(1 - \frac{\cos^{-1}(S_0)}{\pi})}{\log(1 - \frac{\cos^{-1}(cS_0)}{\pi})}. \tag{9}$$

They called their algorithm SIMPLE-LSH as it avoids the parameter tuning process of L2-ALSH. Moreover, SIMPLE-LSH is proved to be a universal LSH for MIPS under any valid configuration of $S_0$ and $c$. SIMPLE-LSH also obtains better (lower) $\rho$ values than L2-ALSH and SIGN-ALSH in theory and outperforms both of them empirically [Shrivastava and Li, 2015].

# 3 Norm-Ranging LSH

In this section, we first motivate norm-ranging LSH by showing the problem of SIMPLE-LSH on real datasets, then introduce how norm-ranging LSH (or RANGE-LSH for short) solves the problem.

## 3.1 SIMPLE-LSH on Real Datasets

We plot the relation between $\rho$ and $S_0$ for SIMPLE-LSH in Figure 1(a). Recall that the query time complexity of SIMPLE-LSH is $\mathcal{O}(n^\rho \log n)$ and observe that $\rho$ is a decreasing function of $S_0$. As $\rho$ is large when $S_0$ is small, SIMPLE-LSH suffers from poor query performance when the maximum inner product between a query and the items is small. Before applying the transformation in (8), SIMPLE-LSH requires the 2-norm of the items to be bounded by 1, which is achieved by normalizing the items with the maximum 2-norm $U = \max_{x \in \mathcal{S}} \|x\|$. Assuming $q^\top x = S$ for item vector $x$, we

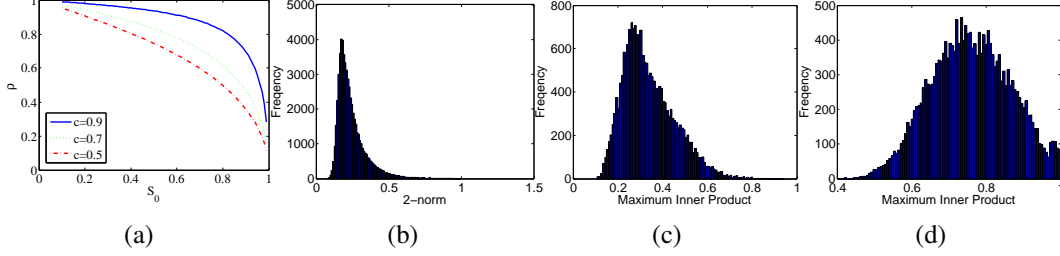

(a)            (b)            (c)            (d)

Figure 1: (a) The relation between $\rho$ and $S_0$; (b) 2-norm distribution of the SIFT descriptors in the ImageNet dataset (maximum 2-norm scaled to 1); (c) The distribution of the maximum inner product of the queries after the normalization process of SIMPLE-LSH; (d) The distribution of the maximum inner product of the queries after the normalization process of RANGE-LSH (32 sub-datasets).

---

**Algorithm 1** Norm-Ranging LSH: Index Building

---

1: **Input:** Dataset $\mathcal{S}$, dataset size $n$, number of sub-datasets $m$
2: **Output:** A hash index $\mathcal{I}_j$ for each of the $m$ sub-datasets
3: Rank the items in $\mathcal{S}$ according to their 2-norms;
4: Partition $\mathcal{S}$ into $m$ sub-datasets $\{\mathcal{S}_1, \mathcal{S}_2, ..., \mathcal{S}_m\}$ such that $\mathcal{S}_j$ holds items whose 2-norms ranked in the range $[\frac{(j-1)n}{m}, \frac{jn}{m}]$;
5: **for** every sub-dataset $\mathcal{S}_j$ **do**
6:     Use $U_j = \max_{x \in \mathcal{S}_j} \|x\|$ to normalize $\mathcal{S}_j$;
7:     Apply SIMPLE-LSH to build index $\mathcal{I}_j$ for $\mathcal{S}_j$;
8: **end for**

---

have $q^\top x = S/U$ after normalization. If $U$ is significantly larger than $\|x\|$, the inner product will be scaled to a small value, and small inner product leads to high query complexity.

We plot the distribution of the 2-norm of a real dataset in Figure 1(b). The distribution has a long tail and the maximum 2-norm is much larger than the majority of the items. We also plot in Figure 1(c) the distribution of the maximum inner product of the queries after the normalization process of SIMPLE-LSH. The results show that for the majority of the queries, the maximum inner product is small, which translates into a large $\rho$ and poor theoretical performance.

The long tail in 2-norm distribution also harms the performance of SIMPLE-LSH in practice. If $\|x\|$ is small after normalization, the $\sqrt{1 - \|x\|^2}$ term, which is irrelevant to the inner product between $x$ and $q$, will be dominant in $P(x) = [x; \sqrt{1 - \|x\|^2}]$. In this case, the result of sign random projection in (4) will be largely determined by the last entry of $a$, causing many items to be gathered in the same bucket. In our sample run of SIMPLE-LSH on the ImageNet dataset [Deng et al., 2009] with a code length of 32, there are only 60,000 buckets and the largest bucket holds about 200,000 items. Considering that the ImageNet dataset contains roughly 2 million items and 32-bit code offers approximately $4 \times 10^9$ buckets, these statistics show that the large $\sqrt{1 - \|x\|^2}$ term severely degrades bucket balance in SIMPLE-LSH. Bucket balance is important for the performance of binary hashing algorithms such as SIMPLE-LSH because they use Hamming distance to determine the probing order of the buckets [Cai, 2016, Gong et al., 2013]. If the number of buckets is small or some buckets contain too many items, Hamming distance cannot define a good probing order for the items, which results in poor query performance.

### 3.2 Norm-Ranging LSH

The index building and query processing procedures of RANGE-LSH are presented in Algorithm 1 and Algorithm 2, respectively. To solve the excessive normalization problem of SIMPLE-LSH, RANGE-LSH partitions the items into $m$ sub-datasets according to the percentiles of the 2-norm distribution so that each sub-dataset contains items with similar 2-norms. Note that ties are broken arbitrarily in the ranking process of Algorithm 1 to ensure that the percentiles based partitioning works even when many items have the same 2-norm. Instead of using $U$, i.e., the maximum 2-norm in the entire dataset, SIMPLE-LSH uses the local maximum 2-norm $U_j = \max_{x \in \mathcal{S}_j} \|x\|$ in each sub-dataset for normalization, so as to keep the inner products of the queries large. In Figure 1(d), we plot the

**Algorithm 2** Norm-Ranging LSH: Query Processing

---

1: **Input:** Hash indexes $\{\mathcal{I}_1, \mathcal{I}_2, ..., \mathcal{I}_m\}$ for the sub-datasets, query $q$
2: **Output:** A $c$-approximate MIPS $x^\star$ to $q$
3: **for** every hash index $\mathcal{I}_j$ **do**
4:     Conduct MIPS with $q$ to get $x_j^\star$;
5: **end for**
6: Select the item in $\{x_1^\star, x_2^\star, ..., x_m^\star\}$ that has the maximum inner product with $q$ as the answer $x^\star$;

---

maximum inner product of the queries after the normalization process of RANGE-LSH. Comparing with Figure 1(c), the inner products are significantly larger. As a result, given $S_0$, the $\rho_j$ of sub-dataset $\mathcal{S}_j$ becomes $\rho_j = G(c, S_0/U_j)$, which is smaller than $\rho = G(c, S_0/U)$ if $U_j < U$. The smaller $\rho$ values translate into better query performance. The idea of dataset partitioning is also used in [Andoni and Razenshteyn, 2015] for $L_2$ similarity search, where the partitioning is conducted in a pseudo-random manner. In the following, we prove that RANGE-LSH achieves a lower query time complexity bound than SIMPLE-LSH under mild conditions.

**Theorem 1.** RANGE-LSH *attains lower query time complexity upper bound than that of* SIMPLE-LSH *for c-approximate MIPS with sufficiently large $n$, if the dataset is partitioned into $m = n^\alpha$ sub-datasets and there are at most $n^\beta$ sub-datasets with $U_j = U$, where $0 < \alpha < \min\{\rho, \frac{\rho - \rho^\star}{1 - \rho^\star}\}$, $0 < \beta < \alpha\rho$, $\rho^\star = \max_{\rho_j < \rho} \rho_j$, $\rho_j = G(c, S_0/U_j)$ and $\rho = G(c, S_0/U)$.*

*Proof.* Firstly, we prove the correctness of RANGE-LSH, that is, it indeed returns a $cS_0$ approximate answer with probability at least $1 - \delta$. Note that $S_0$ is a pre-specified parameter common to all sub-datasets rather than the actual maximum inner product in each sub-dataset. If there is an item $x^\star$ having an inner product of $S_0$ with $q$ in the original dataset, it is certainly contained in one of the sub-datasets. When we conduct MIPS on all the sub-datasets, the sub-dataset containing $x^\star$ will return an item having inner product $cS_0$ with $q$ with probability at least $1 - \delta$ according to the guarantee of SIMPLE-LSH. The final query result is obtained by selecting the optimal one (the one having the largest inner product with $q$) from the query answers of all sub-dataset according to Algorithm 2, which is guaranteed to be no less than $cS_0$ with probability at least $1 - \delta$.

Now we analyze the query time complexity of RANGE-LSH. For each sub-dataset $\mathcal{S}_j$, it contains $n^{1-\alpha}$ items and the query time complexity upper bound of $c$-approximate MIPS is $\mathcal{O}(n^{(1-\alpha)\rho_j} \log n^{1-\alpha})$ with $\rho_j = G(c, S_0/U_j)$. As there are $m = n^\alpha$ sub-datasets, the time complexity of selecting the optimal one from the answers of all sub-datasets is $\mathcal{O}(n^\alpha)$. Considering $\rho_j$ is an increasing function of $U_j$ and there are at most $n^\beta$ sub-datasets with $U_j = U$, the query time complexity of RANGE-LSH can be expressed as:

$$
\begin{aligned}
f(n) &= n^\alpha + \sum_{j=1}^{n^\alpha} n^{(1-\alpha)\rho_j} \log n^{1-\alpha} < n^\alpha + \sum_{j=1}^{n^\alpha} n^{(1-\alpha)\rho_j} \log n \\
&= n^\alpha + \sum_{j=1}^{n^\alpha - n^\beta} n^{(1-\alpha)\rho_j} \log n + n^\beta n^{(1-\alpha)\rho} \log n \\
&< n^\alpha + n^\alpha n^{(1-\alpha)\rho^\star} \log n + n^\beta n^{(1-\alpha)\rho} \log n
\end{aligned}
\tag{10}
$$

Strictly speaking, the equal sign in the first line of (10) is not rigorous as the constants and non-dominant terms in the complexity of querying each sub-dataset are ignored. However, we are interested in the order rather than the precise value of query time complexity, so the equal sign is used for the conciseness of expression. Comparing $f(n)$ with the $\mathcal{O}(n^\rho \log n)$ complexity of SIMPLE-LSH,

$$
\begin{aligned}
\frac{f(n)}{n^\rho \log n} &< \frac{n^\alpha + \left(n^\alpha n^{(1-\alpha)\rho^\star} + n^\beta n^{(1-\alpha)\rho}\right) \log n}{n^\rho \log n} \\
&= n^{\alpha - \rho}/\log n + n^{\alpha + (1-\alpha)\rho^\star - \rho} + n^{\beta - \alpha\rho}
\end{aligned}
\tag{11}
$$

(11) goes to 0 with sufficiently large $n$ when $\alpha \leq \rho$, $\alpha + (1-\alpha)\rho^\star < \rho$ and $\beta - \alpha\rho < 0$, which is satisfied by $\alpha < \min\{\rho, \frac{\rho - \rho^\star}{1 - \rho^\star}\}$ and $\beta < \alpha\rho$. $\qquad\square$

Note that the conditions of Theorem (1) can be easily satisfied. Theorem (1) imposes an upper bound instead of a lower bound on the number of sub-datasets, which is favorable as we usually do not want to partition the dataset into a large number of sub-datasets. Moreover, the condition that the number of sub-datasets with $U_j = U$ is smaller than $n^{\alpha\rho}$ is easily satisfied as very often only the sub-dataset that contains the items with the largest 2-norms has $U_j = U$. The proof also shows that RANGE-LSH is not limited to datasets with long tail in 2-norm distribution. As long as $U > U_j$ holds for most sub-datasets, RANGE-LSH can provide better performance than SIMPLE-LSH. We acknowledge that RANGE-LSH and SIMPLE-LSH are equivalent when all items have the same 2-norm. However, MIPS is equivalent to angular similarity search in this case, and thus can be solved directly with sign random projection rather than SIMPLE-LSH. In most applications that involve MIPS, there are considerable variations in the 2-norms of the items and RANGE-LSH will be beneficial.

The lower theoretical query time complexity of RANGE-LSH also translates into much better bucket balance in practice. On the ImageNet dataset, RANGE-LSH with 32-bit code maps the items to approximately 2 million buckets and most buckets contain only 1 item. Comparing with the statistics of SIMPLE-LSH in Section 3.1, these numbers show that RANGE-LSH has much better bucket balance, and thus better ability to define a good probing order for the items. This can be explained by the fact that RANGE-LSH uses more moderate scaling factors for each sub-dataset than SIMPLE-LSH, thus significantly reducing the magnitude of the $\sqrt{1 - \|x\|^2}$ term in $P(x) = [x; \sqrt{1 - \|x\|^2}]$.

## 3.3 Similarity Metric

Although the theoretical guarantee of LSH only holds when using multiple hash tables, in practice, LSH is usually used in a single-table multi-probe fashion for candidate generation for similarity search [Andoni et al., 2015, Lv et al., 2007]. The buckets(items) are ranked according to the number of identical hashes they have with the query (e.g., Hamming ranking) and the top-ranked buckets are probed first. Multi-probing is challenging for RANGE-LSH as different sub-datasets use different normalization constants and buckets from different sub-datasets cannot be ranked simply according to their number of identical hashes. To support multi-probe in RANGE-LSH, we formulate a similarity metric for bucket ranking that is efficient to manipulate.

Combining the index building process of RANGE-LSH and the collision probability of sign random projection in (4), the probability that an item $x \in \mathcal{S}_j$ and the query collide on one bit is $p = 1 - \frac{1}{\pi} \cos^{-1} \left( \frac{q^\top x}{U_j} \right)$, where $U_j$ is the maximum 2-norm in sub-dataset $\mathcal{S}_j$. Denote the code length as $L$ and the number of identical hashes bucket $b$ has with the query as $l$, we can obtain an estimate of the collision probability $p$ as $\hat{p} = l/L$. Plug $\hat{p}$ into the collision probability, we get an estimate $\hat{s}$ of the inner product between $q$ and the items in bucket $b$ (from sub-dataset $\mathcal{S}_j$) as:

$$\hat{s} = U_j \cos \left[ \pi (1 - \frac{l}{L}) \right]. \tag{12}$$

Therefore, we can compute $\hat{s}$ for the buckets(items) and use it for ranking. When $l > L/2$, $\cos \left[ \pi (1 - \frac{l}{L}) \right] > 0$, thus larger $U_j$ indicates higher inner product while the opposite is true when $l < L/2$. Since the code length is limited and $l/L$ can diverge from the actual collision probability $p$, it is possible that a bucket has large $U_j$ and large inner product with $q$, but it happens that $l < L/2$. In this case, it will be probed late in the query process, which harms query performance. To alleviate this problem, we adjust the similarity indicator to $\hat{s} = U_j \cos \left[ \pi (1 - \epsilon)(1 - \frac{l}{L}) \right]$, where $0 < \epsilon < 1$ is a small number. For the adjusted similarity indicator, $\cos \left[ \pi (1 - \epsilon)(1 - \frac{l}{L}) \right] < 0$ only when $l < L \left[ \frac{1}{2} - \frac{\epsilon}{2(1-\epsilon)} \right]$, which leaves some room to accommodate the randomness in hashing.

Note that the similarity metric in (12) can be manipulated efficiently with low complexity. We can calculate the values of $\hat{s}$ for all possible combinations of $l$ and $U_j$, and sort them during index building. Note that the sorted structure is common for all queries and does not take too much space [3]. When a query comes, query processing can be conducted by traversing the sorted structure in ascending order. For a pair $(U_j, l)$, $U_j$ determines the sub-dataset while $l$ is used to choose the buckets to probe in that sub-dataset with standard hash lookup. We also provide an efficient method to rank the items when code length is large and there are many empty buckets in the supplementary material.

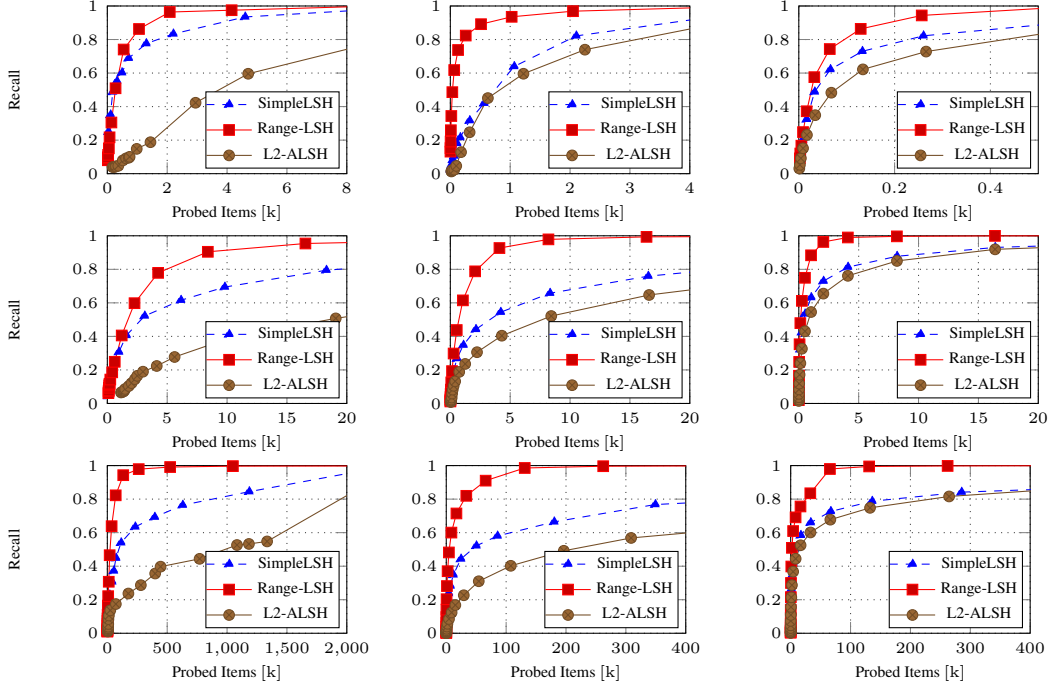

Figure 2: Probed item-recall curve for top 10 MIPS on Netflix (top row), Yahoo!Music (middle row), and ImageNet (bottom row). From left to right, the code lengths are 16, 32 and 64, respectively.

## 4 Experimental Results

We used three popular datasets, i.e., Netflix, Yahoo!Music and ImageNet, in the experiments [4]. For the Netflix dataset and Yahoo!Music dataset, the user and item embeddings were obtained using alternating least square based matrix factorization [Yun et al., 2013], and each embedding has 300 dimensions. We used the item embeddings as dataset items and the user embeddings as queries. The ImageNet dataset contains more than 2 million SIFT descriptors of the ImageNet images, and we sampled 1000 SIFT descriptors as queries and used the rest as dataset items. Note that the 2-norm distributions of the Netflix and Yahoo!Music embeddings do not have long tail and the maximum 2-norm is close to the median (see the supplementary material), which helps verify the robustness of RANGE-LSH to different 2-norm distributions. For each dataset, we report the average performance of 1,000 randomly selected queries.

We compared RANGE-LSH with SIMPLE-LSH and L2-ALSH. For L2-ALSH, we used the parameter setting recommended by its authors, i.e., $m = 3$, $U = 0.83$, $r = 2.5$. For RANGE-LSH, part of the bits in the binary code are used to encode the index of the sub-datasets and the rest are generated by hashing. For example, if the code length is 16 and the dataset is partitioned into 32 sub-datasets, the 16-bit code of RANGE-LSH consists of 5 bits for indexing the 32 sub-datasets, while the remaining 11 bits are generated by hashing. We partitioned the dataset into 32, 64 and 128 sub-datasets under a code length of 16, 32 and 64, respectively. For fairness of comparison, all algorithms use the same total code length. Following existing researches, we mainly compare the performance of the algorithms for single-table based multi-probing. While a comparison of the multi-table single probe performance between RANGE-LSH and SIMPLE-LSH can be found in the supplementary material.

We plot the probed item-recall curves in Figure 2. The results show that RANGE-LSH probes significantly less items compared with SIMPLE-LSH and L2-ALSH at the same recall. Due to space limitation, we only report the performance of top 10 MIPS, the performance under more configurations can be found in the supplementary material.

Recall that Algorithm 1 partitions a dataset into sub-datasets according to percentiles in the 2-norm distribution. We tested an alternative partitioning scheme, which divides the domain of 2-norms into

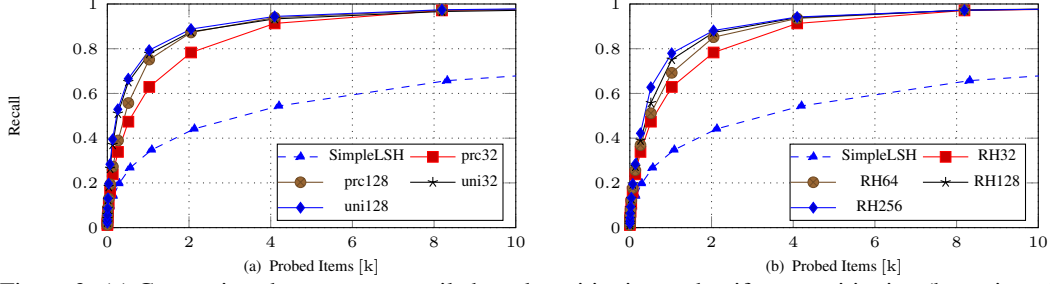

Figure 3: (a) Comparison between percentile based partitioning and uniform partitioning (best viewed in color), the code length is 32 bit and the dataset is Yahoo!Music. prc32 and uni32 mean percentile and uniform partitioning with 32 sub-datasets, respectively. (b) The influence of the number of sub-datasets on the performance on the Yahoo!Music dataset, the code length is 32 bit and the number of sub-datasets varies from 32 to 256. RH32 means RANGE-LSH with 32 sub-datasets.

*uniformly* spaced ranges and items falling in the same range are partitioned into the same sub-dataset. The results are plotted in Figure 3(a), which shows that uniform partitioning achieves slightly better performance than percentile partitioning. This proves RANGE-LSH is general and robust to different partitioning methods as long as items with similar 2-norms are grouped into the same sub-dataset. We also experimented the influence of the number of sub-datasets on performance in Figure 3(b). The results show that performance improves with the number of sub-datasets when the number of sub-datasets is still small, but stabilizes when the number of sub-datasets is sufficiently large.

## 5    Extension to L2-ALSH

In this section, we show that the idea of RANGE-LSH, which partitions the original dataset into sub-datasets with similar 2-norms, can also be applied to L2-ALSH [Shrivastava and Li, 2014] to obtain more favorable (smaller) $\rho$ values than (7). Note that we get (7) from (6) as we only have $0 \le \|x\| \le S_0$ if the entire dataset is considered. For a sub-dataset $\mathcal{S}_j$, if we have the range of its 2-norms as $u_{j-1} < \|x\| \le u_j$, $u_j < S_0$ and $u_{j-1} > 0$, we can obtain the $\rho_j$ of $\mathcal{S}_j$ as:

$$\rho_j = \frac{\log F_r(\sqrt{1 + m/4 - 2U_j S_0 + (U_j u_j)^{2^{m+1}}})}{\log F_r(\sqrt{1 + m/4 - 2cU_j S_0 + (U_j u_{j-1})^{2^{m+1}}})}. \tag{13}$$

As $u_j < S_0$ and $u_{j-1} > 0$, the collision probability in the numerator increases while the the collision probability in the denominator decreases if we compare (13) with (7). Therefore, we have $\rho_j < \rho$. Moreover, partitioning the original dataset into sub-datasets allows us to use different normalization factor $U_j$ (in addition to $m$ and $r$) for each sub-dataset and we only need to satisfy $U_j < 1/u_j$ rather than $U < 1/\max_{x \in \mathcal{S}} \|x\|$, which allows more flexibility for parameter optimization. Similar to Theorem (1), it can also be proved that dividing the dataset into sub-datasets results in an algorithm with lower query time complexity than the original L2-ALSH. We show empirically that dataset partitioning improves the performance of L2-ALSH in the supplementary material.

## 6    Conclusions

Maximum inner product search (MIPS) has many important applications such as collaborative filtering and computer vision. We showed that, SIMPLE-LSH, the state-of-the-art hashing method for MIPS, has critical performance limitations due to the long tail in the 2-norm distribution of real datasets. To tackle this problem, we proposed RANGE-LSH, which attains provably lower query time complexity than SIMPLE-LSH under mild conditions. In addition, we also formulated a novel similarity metric that can be processed with low complexity. The experimental results showed that RANGE-LSH significantly outperforms SIMPLE-LSH, and RANGE-LSH is robust to the shape of 2-norm distribution and different partitioning methods. We also showed that the idea of SIMPLE-LSH hashing is general and can be applied to boost the performance of L2-ALSH. The superior performance of RANGE-LSH can benefit many applications that involve MIPS.

**Acknowledgments**

We thank the reviewers for their valuable comments. This work was supported in part by Grant CUHK 14222816 from the Hong Kong RGC.

## Footnotes

[1]Asymmetric transformation means that the transformations for the queries and the items are different, while symmetric transformation means the same transformation is applied to the items and queries.

[2] $c$-approximate nearest neighbor search solves the following problem: given parameters $S_0 > 0$ and $\delta > 0$, if there exists an $S_0$-near neighbor of $q$ in $\mathcal{S}$, return some $cS_0$-near neighbor in $\mathcal{S}$ with probability at least $1 - \delta$.

[3] $l$ can take $L + 1$ values, $U_j$ can take $m$ values, so the size of the sorted structure is $mL + m$.

[4]Experiment codes https://github.com/xinyandai/similarity-search/tree/mipsex.

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
