[Supplementary Material]

# Supplementary Material for Norm-ranging LSH for Maximum Inner Product Search

## 1 More examples of real datasets with long tails in 2-norm distribution

In this part, we provide 3 more examples of real datasets that have a long tail in their 2-norm distributions in Figure 1. Although we give example using the ImageNet dataset in the paper, we note that the ImageNet dataset is not an outlier and there are many real datasets with long tails in their 2-norm distributions.

Figure 1: More datasets with long tails in their 2-norm distributions. From left to right, the datasets are glove2.2m, nuswide and msong. The maximum 2-norm is normalized to 1.

## 2 Efficient ranking for the similarity metric with long code length

In Section 3.3, we introduced the similarity metric $\hat{s} = U_j \cos\left[\pi(1-\epsilon)(1-\frac{l}{L})\right]$, which can be used to rank the buckets/items when RANGE-LSH is used for single table based multi-probing. When the code length is not large, we can sort all possible combinations of $l$ and $U_j$ in the index building phase and generate the bucket to probe by traversing the sorted structure in ascending order. However, when the code length is long and the number of available buckets is much larger that the number of items, generate-to-probe is not efficient and many empty buckets will be generated. The more efficient solution is to rank the items according to their similarity metric. In this part, we provide an efficient algorithm for ranking the items according to our similarity metric.

The algorithm is conducted in 2 phases: 1) Start with the sub-dataset with the maximum 2-norm and scan the hash index of the sub-datasets by calculating the Hamming distance between the items and the query, if the hamming distance between query $q$ and item $x$ is $l$, put $x$ into the $l$-th queue along with its sub-dataset index $j$; 2) If the code length if $L$, we have $L+1$ queues, conduct a merge sort on the $L+1$ queues according to the value of the similarity metric.

Note that the values of the similarity metric do not need to be calculated in query processing. The similarity metric values for all possible combinations of $l$ and $U_j$ can be precomputed and accessed with lookup. Moreover, we can also generate the item to probe in an on demand manner with the algorithm in the second phase and do not need to actually complete the merge sort.

# 3 The 2-norm distribution of the datasets used in the experiments

In the paper, we motivate our RANGE-LSH with the long tail in the 2-norm distribution, e.g., that in the ImageNet dataset. However, the proof of Theorem 1 shows that RANGE-LSH is actually more general and can outperform SIMPLE-LSH as long as there are not too many sub-datasets having a maximum 2-norm equal to the global maximum in the entire dataset. In Figure 2, we show the 2-norm distributions of the item embeddings obtained via matrix factorization on the Netflix dataset and Yahoo! Music dataset along with the SIFT descriptors from the ImageNet dataset. The Netflix dataset and Yahoo! Music dataset do not have a long tail in 2-norm distribution and the median is close to the maximum. However, RANGE-LSH also significantly outperforms SIMPLE-LSH and L2-ALSH on these two datasets as reported in the main paper. Therefore, RANGE-LSH is robust to different 2-norm distributions and can handle a wider variety of real datasets.

Figure 2: The 2-norm distribution of the datasets used in the experiments, the maximum 2-norm is normalized to 1. From left to right, the datasets are Netflix, Yahoo! Music and ImageNet.

# 4 Multi-table performance comparison

Figure 3: Number of returned candidates comparison between SIMPLE-LSH and RANGE-LSH when using multiple hash table.

LSH based algorithms are also commonly used for multi hash table based single probe, in which $T$ hash tables are generated and each hash table contains the entire dataset. For each hash table, only the bucket that has the same code as the query is probed. To compare the multi-table performance of SIMPLE-LSH and RANGE-LSH, we followed [Andoni et al., 2015] and set the parameters such that the empirical probability of finding the maximum inner product is at least 0.9. We do not compare with other algorithms as SIMPLE-LSH is shown to outperform them. Figure 7 reports that the number of candidates returned by RANGE-LSH is an order of magnitude fewer than SIMPLE-LSH. The main reason that RANGE-LSH generates less candidates is because it is able to use less hash table to achieve the same recall.

Figure 4: Recall versus the number of probed items (best viewed in colors) for the top 1 MIPS on Netflix (top row), Yahoo!Music (middle row), and ImageNet (bottom row). From left to right, the code lengths are 16, 32 and 64.

# 5 More experimental results

In this part, we provide more experimental results on the Netflix, Yahoo! Music and ImageNet datasets under other configurations of $k$.

## 5.1 Top 1 MIPS

We report the performance of RANGE-LSH, SIMPLE-LSH and L2-ALSH for the top 1 MIPS in Figure 4. The results show that RANGE-LSH significantly outperforms SIMPLE-LSH and L2-ALSH for the top 1 MIPS.

Figure 5: Recall versus the number of probed items (best viewed in colors) for the top 20 MIPS on Netflix (top row), Yahoo!Music (middle row), and ImageNet (bottom row). From left to right, the code lengths are 16, 32 and 64.

## 5.2 Top 20 MIPS

We report the performance of RANGE-LSH, SIMPLE-LSH and L2-ALSH for the top 20 MIPS in Figure 5. The results show that RANGE-LSH significantly outperforms SIMPLE-LSH and L2-ALSH for the top 20 MIPS.

Figure 6: Recall versus the number of probed items (best viewed in colors) for the top 50 MIPS on Netflix (top row), Yahoo!Music (middle row), and ImageNet (bottom row). From left to right, the code lengths are 16, 32 and 64.

## 5.3 Top 50 MIPS

We report the performance of RANGE-LSH, SIMPLE-LSH and L2-ALSH for the top 50 MIPS in Figure 5. The results show that RANGE-LSH significantly outperforms SIMPLE-LSH and L2-ALSH for the top 50 MIPS.

From the figures, we can conclude that the performance improvement of RANGE-LSH over SIMPLE-LSH and L2-ALSH is consistent over different configurations of $k$.

# 6 Performance improvements on L2-ALSH

Figure 7: Effect of dataset partitioning on L2-ALSH.

In this part, we show the performance improvement of dataset partitioning on L2-ALSH. We name the dataset partition version of L2-ALSH as Range-ALSH. Similar to RANGE-LSH, Range-ALSH also estimates the inner product based on the number of identical hashes and the collision probability, and uses the estimated inner product to rank the buckets/items. We compare Range-ALSH with L2-ALSH on the ImageNet dataset for top 10 MIPS under a code length of 32 in Figure **??**. The result shows that dataset partitioning also improves the performance of L2-ALSH.

## References

A. Andoni, P. Indyk, T. Laarhoven, I. P. Razenshteyn, and L. Schmidt. Practical and optimal LSH for angular distance. In *NIPS*, pages 1225–1233, 2015.