[Reviews · NeurIPS 2018]

Reviewer 1



The paper shows a new reduction from the Maximum Inner Produce Search problem to the Nearest Neighbor Search problem (NIPS). Basically, the idea is to group dataset points based on the length, and then within each bucket building an LSH data structure with separate parameters. I like that this simple idea works decently in practice, however I have a few (somewhat serious) reservations about the paper. 1. First, this idea was known and used before, see, e.g., ProjectBall procedure in Figure 3 of https://arxiv.org/pdf/1501.01062.pdf . Definitely, this paper must be cited. On a related note, it's unacceptable not to cite any LSH paper beyond the usual Indyk Motwani reference (in particular, for multi-probe indexing scheme, one should cite https://arxiv.org/abs/1509.02897 ). 2. The paper claims "We prove that NORM - RANGING LSH has lower query time complexity than SIMPLE - LSH", and then "We acknowledge that RANGE - LSH and SIMPLE - LSH are equivalent when all items have the same 2-norm. " I think this is extremely poor taste, and unacceptable scholarly standard. 3. The experiments can be done better. The authors just test optimized linear scan based on compact sketches instead of proper sub-linear time retrieval that is analyzed before. I think it's a bit weird to analyze one algorithm and then test another one.

Reviewer 2



The manuscript clearly motivates the issue with the state-of-the-art solution for approximate MIPS, and proposes a very intuitive idea to improve the performance of existing methods for approximate MIPS. The theoretical and the thorough empirical results clearly demonstrate the superiority of the proposed solution relative to the considered baselines. The authors explain the cost of reducing MIPS to NNS via simple-lsh -- the squashing of the maximum inner-products when the 2-norms distribution of the points have a long tail. The manuscript presents the theoretical and practical consequences of this issue. The authors bypass this issue by splitting the dataset into subsets and solving MIPS in each of the subset with simple-lsh. This new scheme is guaranteed to be theoretically faster than simple-lsh on the whole dataset (under some regularity conditions). The authors also provide a technique to handle the practical issue of working with multiple hash indices obtained from having to create an index for each of the subsets of the dataset, which allows the use of schemes like multi-probe. The proposed technique is thoroughly evaluated in the empirical section; the authors present results both in terms of the actual runtimes (which is the quantity that matters at the end) as well as the number of points probed (which clearly demonstrates the gains obtained only by the algorithm and not by implementation specifics). The authors evaluate various aspects of the proposed solution and the results are extremely positive. In my opinion, this paper presents a technique to improve the performance of existing MIPS solutions in a clear manner, addressing every aspect of the proposed technique. I do not have any comments/questions/concerns with the proposed technique in this manuscript. The only concern I have is with the premise of the paper that simple-lsh is the state-of-the-art when there are published results (see [A,B]) which have significantly improved upon the performance of simple-lsh. [A] Christina Teflioudi and Rainer Gemulla. Exact and approximate maximum inner product search with lemp. ACM Transactions on Database Systems (TODS), 42(1):5, 2017. [B] Omid Keivani, Kaushik Sinha, and Parikshit Ram. Improved maximum inner product search with better theoretical guarantees. In Neural Networks (IJCNN), 2017 International Joint Conference on, pages 2927–2934. IEEE, 2017.

Reviewer 3



Authors find that the inner-product-to-cosine-reduction technique by Neyshabur and Srebro suffers from a disbalance in the L2-norm distribution. They show that this can be improved by splitting the data set (plus some additional improvements). They present a proof that such a split has theoretical guarantees. In addition, they proposed an approach to improve probing order of the buckets in their norm-split index. They demonstrate their methods is partly applicable to A2-LSH, but it will likely improve other SOTA methods. This is very solid work: I would vote for acceptance despite it misses comparison to some state-of-the-art and uses mostly small data sets (even ImageNet is not really large-scale as authors claim). PROS: This is a well-written paper: I enjoyed reading it. Authors make an interesting finding that can potentially benefit several methods for k-NN search using the maximum-inner product as the similarity function. In particular, the method of Andoni et al [2015]. Possibly, data set splitting can be useful even for graph-based retrieval methods (despite some of your tricks are applicable to random-projection LSH only). They present several other interesting findings, some of which I briefly mention in detailed comments. CONS: Authors do not compare against state-of-the art graph-based algorithms (Malkov and Yashunin ). Evaluation leaves somewhat to be desired, because two out of three data sets are small (Netfix is in fact tiny). ImageNet is not very big either. Additional baseline that would be nice to run: 1) Andoni et al method for the cosine similarity search combined with the inner-product-to-cosine-reduction by Neyshabur and Srebro. One should be quite curious if Andoni et al would benefit from your data set splitting approach. Again, I understand it wouldn't be exactly straightforward to apply: yet, even a simple data set splitting with parallel searching over split may be beneficial. I think this would be a strong baseline as Andoni et all improve over simple-LSH quite a bit. 2) My feeling is that your extension to A2-LSH (Section 5) should have been implemented and tested. SOME FURTHER COMMENTS: 61-62: Wasn't the original definition specified in terms of the distance? Does it make a difference when you reformulate it using the similarity instead? I think it would beneficial to formally introduce similarity and explain how it relates to the distance (e.g., there's an obvious connection for the cosine similarity). 68-69: the probability GIVEN THAT A RANDOM FUNCTION is drawn randomly and independently FOR EACH PAIR of vectors. This nearly always silent assumption of the LSH community was deservedly criticized by Wang et al. I think there is nothing bad about using this simplified assumption, but it is quite bad NOT TO MENTION it. Nearly nobody gets this subtlety and I have seen a machine learning professor from a top university baffled by the question what is actually assumed to be random in the theory of LSH. 103: Complexity under which assumptions? Also, missing citation to back up this. 108: It would be nice to add: i.e., the higher is the similarity, the lower is the complexity. Otherwise: nice observation also backed up by theory! 117-118: Nice observation too! 226: 2 million is not large at all! And this is your only reasonably large data set. REFERENCES Andoni, Alexandr, et al. "Practical and optimal LSH for angular distance." Advances in Neural Information Processing Systems. 2015. Malkov, Yu A., and D. A. Yashunin. "Efficient and robust approximate nearest neighbor search using Hierarchical Navigable Small World graphs." arXiv preprint arXiv:1603.09320 (2016). Wang, Hongya, et al. "Locality sensitive hashing revisited: filling the gap between theory and algorithm analysis.